# The Planar Core–Shell Junctionless MOSFET

**DOI:** 10.3390/mi16040418

**Published:** 2025-03-31

**Authors:** Cunhua Dou, Weijia Song, Yu Yan, Xuan Zhang, Zhiyu Tang, Xing Zhao, Fanyu Liu, Shujian Xue, Huabin Sun, Jing Wan, Binhong Li, Yun Wang, Tianchun Ye, Yong Xu, Sorin Cristoloveanu

**Affiliations:** 1Guangdong Greater Bay Area Institute of Integrated Circuit and System, Guangzhou 510300, China; 2023221101@njupt.edu.cn (C.D.); song.weijia@uclouvain.be (W.S.); 2022020405@njupt.edu.cn (Y.Y.); zhangxuan@giics.com.cn (X.Z.); 1023223205@njupt.edu.cn (Z.T.); xueshujian@giics.com.cn (S.X.); libinhong@ime.ac.cn (B.L.); wangyun@giics.com.cn (Y.W.); tcye@ime.ac.cn (T.Y.); 2School of Integrated Circuit Science and Engineering, Nanjing University of Posts and Telecommunications, Nanjing 210003, China; hbsun@njupt.edu.cn; 3Institute of Microelectronics of the Chinese Academy of Sciences, Beijing 100029, China; zhaoxing@ime.ac.cn (X.Z.); liufanyu@ime.ac.cn (F.L.); 4School of Information Science and Technology, Fudan University, Shanghai 200433, China; jingwan@fudan.edu.cn

**Keywords:** junctionless, core–shell, mobility, FD-SOI, miniaturization, planar MOSFET

## Abstract

The core–shell junctionless MOSFET (CS-JL FET) meets the process requirements of FD-SOI technology. The transistor body comprises a heavily doped ultrathin layer (core linking the source and the drain), located underneath an undoped layer (shell). Drain current, transconductance, and capacitance characteristics demonstrate striking performance improvement compared with conventional junctionless MOSFETs. The addition of the shell results in one order of magnitude higher mobility (peak value), transconductance, and drive current. The doping and thickness of the core can be engineered to achieve a positive threshold voltage for normally-off operation. The CS-JL FET is compatible with back-biasing and downscaling schemes. The physical mechanisms are revealed by emphasizing the roles of the main device parameters.

## 1. Introduction

The majority of advanced MOS transistors (FinFETs, FD-SOI, nanowires, nanosheets) feature an undoped body and operate in a full depletion regime [1,2,3]. However, heavily doped junctionless transistors (JLs) have the merit of eliminating the burden of junction processing, a key constraint while fabricating monolithic 3D circuits that require a low thermal budget [4]. The JL-FET invented by Colinge [5,6,7] is ultrathin in order for its gate to be able to deplete the body and switch the current off. A very high doping from source to body and drain is instrumental for achieving reasonable drive current and avoiding excessive series resistance.

The unrivaled easiness of the JL-FET comes with intrinsic penalties. Very poor carrier mobility, random doping fluctuations, enhanced noise and impact ionization, and normally-on operation are inevitable consequences of heavy doping. Reducing the body doping and adding extra dopants in the source and drain is not an option since the device is no longer ‘junctionless’.

The core–shell (CS) architecture was recently proposed to eliminate JL drawbacks and increase the performance up to FD-SOI MOSFET level [8,9]. Wrapping an undoped shell around a regular JL core renders the transistor normally-off with high current and mobility. The device was initially conceived as a Gate-All-Around (GAA) nanowire, where the electrostatic control of the gate over the channel is ideal. Since the processing of a GAA core–shell structure is still challenging, we envisioned the implementation of this transistor in state-of-the-art FD-SOI technology [3].

Although the core–shell designation is fit for GAA nanowires, we preserved it for simplicity as the fundamental mechanisms are unchanged. Figure 1 shows that only the upper face of the core is surrounded by the shell. Still, the original concept remains: the drain current combines the contributions of slow electrons that result from the core doping with fast electrons attracted by the gate in the shell.

The planar embodiment of the CS-JL FET (Figure 1) is rather similar to the A2RAM structure operated as a single-transistor, floating-body 1T-DRAM memory [10,11,12,13,14,15,16,17,18,19,20,21]. The idea is to store holes in the undoped shell and measure their impact on the current flowing in the N+ core, also named ‘bridge’ [10,11,12,13,14]. Several generations of A2RAM have been fabricated, and the performance is promising [15,16,17,18,19,20,21]. The CS-JL FET being intended for logic applications, its structure differs in several respects: (a) the shell and core layers are far thinner, (b) the core is more heavily doped, and (c) the junctions have the same doping concentration as the core without receiving additional implants or in situ doping.

Our preliminary results, reported in [22], are here enriched with detailed discussions of the operation mechanisms, performance, and structure optimization. After presenting the device configuration in Section 2, we discuss in Section 3 the electron profiles across the body, before and after the filling of the core and shell. This will clarify the two-step build-up of the charge. The gate-to-channel capacitance characteristics introduced in Section 4 differ from those of regular MOSFETs. They enable determining the total amount of mobile charge, as well as the threshold voltages for core and shell activation. The drain current and transconductance characteristics of CS-JL FETs with variable core doping and thickness are compared to those of classical JL and FD-SOI transistors in Section 5. It is the disparity in carrier mobility between the core and shell that fuels interest in CS-JL FETs. Short-channel effects are examined in Section 6. Section 7 shows the back-gate voltage, which can be used to tune the device performance, including the series resistance impact.

## 2. Architecture of the CS-JL Transistor

For perfect compatibility with the 22 nm node FD-SOI process, our CS-JL devices are all-silicon with the same body thickness (7 nm) as a regular MOSFET. The core situated above the buried oxide (BOX) is typically 3 nm thick with a doping concentration *N_D_* above 10^19^ cm^−3^. The shell is undoped with a residual acceptor-type doping of *N_A_* = 10^15^ cm^−3^, common for FD-SOI wafers. Formed on top of the core, the shell benefits from the gate action. The reciprocal CS-JL configuration with the core above the shell is not envisaged because the shell filling would require operation from the back-gate.

For the benchmark, we also consider FD-SOI MOSFET with 7 nm body thickness and pure JL FETs with the same doping as the core and a thickness of 7 nm or 3 nm (like the core). The metal gate stack features a silicon-midgap work-function (5.1 eV) and a 1 nm equivalent oxide thickness. Raised source and drain are implemented for reducing the parasitic series resistance. The BOX is 15 nm thick, and N^+^ or P^+^ ground-planes with 2 × 10^19^ cm^−3^ doping concentration are used for back-gate biasing.

The upper limits of the core doping and thickness are restricted by the necessity to achieve full depletion in off-state; with a partially depleted core, the device cannot be turned off and loses interest.

Sentaurus TCAD (version 2022) tools were used for solving the 2D Poisson equation and simulating capacitance and current characteristics. The electron mobility is formulated as a function of doping and electric field, according to [9,23,24]. Quantum corrections have a minor impact on the overall characteristics: The electron transport in the core is hardly affected, whereas the threshold voltage of the undoped shell is marginally increased. Carrier profiles subject to quantum confinement are shown in Figure 2. We first assume long-channel devices operated at low drain voltage (*V_D_* = 50 mV) and room temperature.

As far as the technology is concerned, the A2RAM was successfully fabricated with a *shell-first* process [15,16,17]. The shell was nothing but the undoped body of a typical FD-SOI MOSFET. The core was implemented subsequently via ion implantation. A *core-first* alternative is equally feasible: the silicon film of SOI wafer is heavily doped to form the core. Local etching opens a cavity where the shell is epitaxially grown. This solution enables the mixing of different materials, for example, an Si core and an SiGe shell.

## 3. Gate-Induced Charge Distribution

The in-depth electron profiles reproduced in Figure 2a reveal two distinct populations, sequentially activated. With the gate negatively biased (*V_G_* < 0), the body is fully depleted, and no current flows in either the core or in the shell. In the subthreshold region, electrons start to populate the bottom of the core. Their density increases exponentially with *V_G_* until the threshold voltage of the core is reached. This threshold voltage *V_T_* actually holds for the whole transistor. It is worth noting that at threshold, the electron concentration in the core is a small fraction of the nominal doping and not zero as assumed in most JL models.

In this first phase, the role of the gate is to gradually fill the core by starting from the BOX interface up and reducing the depletion depth existing at the core–shell boundary. The electron concentration tends to reach a maximum value imposed by the nominal doping. During this core filling period (*V_G_* > *V_T_*), there are barely a few electrons in the shell, mostly located at the shell-core interface.

In a second phase, further raising the gate voltage forces electrons to accumulate at the surface of the shell as in any MOS transistor. The threshold voltage of the shell (*V_G_ = V_TS_*) defines the transition from weak to strong inversion. As long as the surface layer is in weak or moderate inversion, the replenishment of the core continues. It stops, however, in strong inversion when the carrier distribution within the core becomes rather rigid, screened from the gate action. This may happen even if the total core charge has not attained the maximum value expected from doping (*N_D_* × *T_core_*), where *T_core_* is the thickness of the core layer. This is particularly the consequence of a negative back-gate voltage or a P-type ground-plane, which generates depletion at the core–BOX interface (Figure 2b). Selecting a P-type rather than an N-type ground-plane is roughly equivalent to −1 V back bias.

Any increment in *V_G_* > *V_TS_* is absorbed by the shell population. Being unlimited, the inversion charge in the shell increases linearly with the gate voltage overdrive, *Q_inv_ = C_ox_* (*V_G_* − *V_TS_*), with *C_ox_* being the gate oxide capacitance per unit area. In strong inversion, the shell can accommodate more electrons than the core.

The buildup of the shell population with high carrier mobility is the fundamental mechanism in CS-JL FET, unavailable in regular JL. In addition, the effective thickness of the channel increases from the core size in phase 1 to the full body thickness in phase 2.

The mobile charge is obtained by integrating the carrier profile in the core and shell, respectively. The two-phase charge buildup is clearly evidenced in Figure 3a. The arrows indicate the activation of the two channels. In this case of relatively high doping, the two threshold voltages are clearly separated, e.g., *V_TS_* − *V_T_* = 0.67 V here. The core filling is a slower process, which gives rise to a double-slope characteristic.

Figure 3b compares CS-JL, JL, and FD-SOI transistors. The classical JL exhibits a negative threshold voltage, which translates in normally-on operation with a significant charge at *V_G_ =* 0. A thinner JL (3 nm as the core) would have a positive *V_T_* at the cost of very poor current drive. This inconvenience of JL underlines the advantage of the core–shell architecture with normally-off operation.

Systematic simulations conclude that the threshold voltage of CS-JL is defined by the parameters of the core. Reducing the thickness or doping of the core shifts *V_T_* to more positive values. The apparent loss in mobile charge is compensated by the gate, which enriches the shell with the demanded number of electrons.

Replotting the curves in Figure 3b in semi-log scale indicates that in the subthreshold regime the characteristics are all parallel with an ideal swing of 2.3 kT/q per decade of charge (60 mV/decade at room temperature).

## 4. Capacitance Characteristics

The carrier profiles are not experimentally accessible. In practice, it is the gate-to-channel capacitance *C_gc_ = dQ*/*dV_G_* that provides the mobile charge. The split-CV method consists of measuring *C_gc_* between the gate and the interconnected source and drain terminals. Being the series combination of the gate oxide capacitance *C_ox_* and mobile charge capacitance *C_inv_*, *C_gc_* increases from zero in full depletion to an asymptotic value *C_ox_* independent of the transistor configuration (Figure 4). Integration of *C_gc_* from a negative value to *V_G_* yields the charge as a function of gate bias exactly as in Figure 3b.

The capacitance behavior in CS-JL differs qualitatively from the monotonic variation usual in FD-SOI MOSFETs [3]. A shoulder develops reflecting the voltage difference between the early activation of the core preceding that of the shell. Such unusual CV curves have actually been measured in A2RAMs [20,21]. The capacitance corresponding to the shoulder plateau (*C_pl_*) is the series association of *C_ox_* with the capacitance of the fully depleted shell *C_sh_ = ε_si_*/*t_sh_*, where *t_sh_* is the shell’s thickness. It can be used to determine experimentally the shell thickness and core doping [20]. Obviously, there is no shoulder in the FD-SOI capacitance curve (Figure 4).

In CS-JL, the capacitance curves are laterally shifted according to the core doping and core thickness. For increased doping, the curves are moved to the left (Figure 5a), denoting the difficulty of the gate to deplete the core.

It is well documented that the threshold voltage is defined by the inflection point of the capacitance curve, which is the peak of the second derivative of mobile charge [25]. This coincides with the peak of the capacitance derivative *dC_gc_*/*dV_G_*. Interestingly, the shoulder in CS-JL capacitance generates two peaks that account for the sequential activation of two conduction mechanisms, in core and shell (Figure 5b). Double-peak CV curves have been recorded in A2RAM [20,21]. While the left-hand peak is core-dependent, the position of the shell-related peak is rigid, hardly affected by the core properties. The threshold voltage of the shell is always positive and matches the value indicated by the single peak existing in FD-SOI.

The gap between core and shell activation shrinks as the core doping decreases. For doping below 5 × 10^18^ cm^−3^, the two peaks tend to coincide as shown in Figure 5b. This means that the core turns on just before the shell does it. Obviously, the shell cannot turn on before the core; otherwise, the core activation would be blocked by the buildup of the inversion layer.

## 5. Drain Current, Transconductance, and Mobility

### 5.1. Transfer Characteristics

The transfer characteristics in Figure 6 emphasize two striking merits of the CS-JL transistor:(i)In conventional JL, the threshold voltage is definitely negative. CS-JL architecture succeeds in shifting *V_T_* from the unsuitable negative value (normally-on) to a positive value (normally-off).(ii)The drive current is significantly improved (Figure 6a).

Since the traps at the front and back interfaces of the body have been ignored, it is no surprise that all subthreshold curves in Figure 6b feature a quasi-ideal swing. More interestingly, the activation of the shell conduction results in a sudden burst of drain current (hump for *V_G_* ≈ 0.8 V), enabling the CS-JL to match the FD-SOI current and outscore the regular JL by one decade. Such a hump, induced by the ‘shell’ turn-on, has experimentally been monitored in A2RAM transistors [16].

The drain current is the byproduct of carrier charge and velocity. The comparison of charge (Figure 3b) and current (Figure 6a) characteristics confirms that the root of the excellent performance of CS-JL transistors is the carrier velocity impersonated by the effective mobility. Not only are additional electrons accumulated in the shell, but they also flow with much higher velocity. Figure 3b shows that the rate of change of the mobile charge with gate voltage is similar in JL and CS-JL transistors. However, the message from Figure 6a is different: the rate of change of drain current with *V_G_* is much higher in CS-JL.

This difference is summarized by the transconductance variation *g_m_* (*V_G_*) in Figure 7a. The transconductance of the CS-JL FET is initially small and slowly varies with gate voltage, exactly as in the JL transistor. A tremendous boost is visible as soon as the shell starts conducting. Such extraordinary gain by one order of magnitude is produced by fast-moving electrons in the shell.

Figure 7b shows the peaks of the transconductance derivative (second derivative of the drain current). It is well accepted that the position of these peaks marks the threshold voltage value [25]. The JL transistor features two peaks, the left one indicating the threshold voltage (when the body quits full depletion) and the right one the flat-band voltage beyond which an accumulation channel adds at the oxide-body interface.

Two peaks are equally visible in CS-JL, but they carry different information. The left-hand peak does correspond to the core activation, which occurs at higher *V_T_* because the core is thinner than the JL. The right-hand peak shows the shell activation and is related to the MOS stack rather than to the core parameters. In Figure 8c, the currents in the shell and core have been calculated separately in order to underline the meaning of the two threshold voltages.

An intriguing aspect is the threshold voltage of the shell being lower when defined in terms of current (Figure 7b) rather than charge (Figure 5b). The difference (around 100 mV) is again attributable to fast electrons, which make the current onset visible before the charge onset. The charge is a transport-free replica of drain current, exactly as the gate-to-channel capacitance is for the transconductance. Choosing charge or current criteria for *V_T_* is a matter of circuit-designer appreciation for the best fit of *I_D_* (*V_G_*, *V_D_*) characteristics.

In summary, the performance of the CS-JL transistor outscores the JL device and competes with FD-SOI MOSFETs where contacts need to be duly fabricated.

### 5.2. Carrier Mobility

The transconductance mirrors the specifics of the carrier mobility. The *effective* mobility is evaluated by combining the variations of mobile charge (from Split-CV curves in Figure 3, Figure 4 and Figure 5) and drain current (Figure 6).

Figure 8a confirms the great disparity in carrier mobility between our various devices. In highly doped JL transistors, the carriers experience intense Coulomb scattering on ionized impurities, which renders their mobility very poor (50 cm^2^/Vs or less). The impact of gate voltage is marginal. Being undoped, the FD-SOI MOSFET features much higher electron mobility dominated by phonon scattering. It peaks up at 700 cm^2^/Vs before falling in strong inversion as a result of high vertical field and surface scattering.

The CS-JL cumulates both trends. The major weakness of the JL MOSFET in terms of degraded mobility also applies to the core of the CS-JL device. The carrier mobility remains low before being remarkably boosted by the activation of the shell. The carrier transport in undoped shell is virtually free from Coulomb scattering at room temperature, being dominated by phonon scattering and, at high field, by surface roughness scattering. The carrier velocity is equally high in the shell and in the FD-SOI body. In Figure 8a, the electron mobility is an in-depth average that weights the contributions of the core and shell; this is why it looks apparently inferior to the peak mobility in FD-SOI.

The actual value of the carrier mobility in the shell can be extracted with the Y-function [26]. Since this method cannot address two parallel channels, the current in the shell has to be isolated from the total current by subtracting the core current: *I_shell_* = *I_total_* − *I_core_*. The Y-function yields the following:(1)$$Y\left(V_{G}\right)=\frac{I_{shell}}{\sqrt{g_{m,shell}}}=\sqrt{\frac{\mu_{0,shell}C_{ox}WV_{D}}{L}}\left(V_{G}-V_{T,shell}\right)$$
where *μ*_0,*shell*_ is the zero-field mobility in the shell, *W* is the gate width and *L* is the gate length. A linear variation with *V_G_* is obtained (Figure 8b): the slope gives the mobility and the intercept the threshold voltage. The mobility attenuation factor, dominated by the series resistance, can be extracted by plotting *1/√g_m_* versus *V_G_* [3].

A dilemma is the choice of the gate voltage beyond which the current is constant in the core and increases only in the shell. In Figure 8b, we considered that the shell threshold voltage is the mark where the shell current is zero and the core current has reached a maximum value. The Y-function yields a low-field mobility of 530 cm^2^/Vs and a threshold voltage of 0.9 V in agreement with the second peak in Figure 7b.

However, this is a rough approximation because, at shell threshold, the shell current exists already, and the core current continues to increase slowly with *V_G_*. Calculating the exact shell current as a function of *V_G_* (Figure 8c) leads to a superior mobility (640 cm^2^/Vs), which matches that in FD-SOI. The precise separation of core and shell currents, illustrated in Figure 8c, is feasible in TCAD only. In practice, the Y-function is still useful to evaluate the mobility; even with an underestimated value, it underlines the outstanding difference between core and shell mobilities.
Figure 8(**a**) Effective mobility of electrons versus gate voltage in planar CS-JL, conventional JL, and FD-SOI transistors. Same devices as in Figure 6. (**b**) Example of Y-function. (**c**) Separation of core and shell currents.
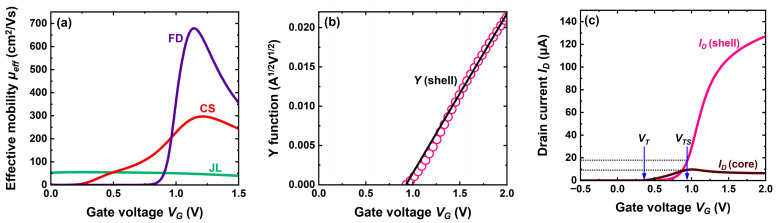



The sound improvement from JL to CS-JL transistors, essentially due to the difference in carrier transport mechanisms, has already been revealed in Gate-All-Around (GAA) CS-JL nanowires [9]. It follows that the generic merits of the core–shell structure are maintained in the planar version, which has the key advantage of full compatibility with FD-SOI fabrication schemes.

### 5.3. Impact of Core Doping

The core doping stands as a very sensitive parameter that controls the threshold voltage, drive current, average mobility, and series resistance. Increasing the doping concentration substantially shifts the characteristics to the left while marginally improving the current (Figure 9a). A relatively small change from 2 × 10^19^ cm^−3^ to 3 × 10^19^ cm^−3^ is able to reduce the threshold voltage by more than 0.3 V, rendering it negative (Figure 9b). In the meantime, the core contribution to total current increases at the expense of degraded transconductance peak (Figure 10a) and average mobility.

In this high-doping range, the threshold voltage varies linearly with *N_D_*, whereas the shell activation is doping-independent (inset Figure 9b). The transition from core conduction to shell activation is clearly marked by a hump in current (Figure 9b), a plateau in transconductance (Figure 10a), and a left-hand peak in the transconductance derivative (Figure 10b).

A lower doping shifts the curves towards normally-off operation and, in very short-channel devices, increases the transconductance peak. For *N_D_* ≤ 10^19^ cm^−3^ the delay between core and shell filling is reduced, which erases the left-hand peak (Figure 10b). An evident penalty is the series resistance degradation.

These trade-off considerations make us conclude that a core doping higher than 2 × 10^19^ cm^−3^ is unsuitable.

### 5.4. Impact of Thickness

Figure 11 confirms that the core parameters are prevalent in the device performance, essentially in weak inversion. A thicker core tends to shift the characteristics towards normally-on operation. With only a 1 nm thicker core (C4S3 versus C3S4 in Figure 11b), the threshold voltage is lowered by 0.3 V, and the gap between core and shell activation is enlarged. The enhanced contribution of a thicker core to the total current is visible only at low gate voltage. In on-state, the benefit is negligible because the current is dominated by the shell. Instead, the off-state current is exponentially degraded by the shifted *V_T_*.

By comparison, the shell thickness has a minor effect. A thinner shell barely increases the threshold voltage (C3S3 versus C3S4 in Figure 11b); the gate being closer to the core, its action is accentuated.

Remark that in on-state (Figure 11a), all these CS-JL devices with conservative body thickness (6–7 nm) are rather equivalent.

## 6. Short-Channel Effects

Figure 12 shows no trace of short-channel effects (SCEs) for channel lengths down to 50 nm. For shorter the gate, all devices show a normal degradation [22]. The threshold voltage roll-off and mobility collapse are both due to the high electric field that develops at the ‘junction’ between the neutral region of terminals and the fully depleted body [27]. This effect is present even in junctionless and core–shell transistors.

In ultrathin SOI transistors, DIBL is essentially due to the fringing electric fields from the source and drain into the BOX, which raise the potential at the back interface exactly as a positive back-gate voltage would do. This Drain-Induced Virtual Substrate Biasing [28] acts directly on the bottom of the body, assisting it to quit full depletion and thus lowering *V_T_*. JL devices appear to be more vulnerable to fringing fields and DIBL than FD-SOI and CS-JL. It is worth noting that the fringing field effect stops once the bottom of the core becomes neutral and prevents further penetration of the field into the body.

The resilience of CS-JL transistors to SCE is in-between that of JL and FD-SOI counterparts. In 20 nm CS-JL, the subthreshold swing degradation is limited to 90 mV/decade (Figure 12a), and DIBL is maintained around 100 mV/V (Figure 12b). The mobility collapse in very short channels is similar in CS-JL and FD-SOI transistors [22].

Increasing the shell thickness from 3 nm to 10 nm is counterproductive: (i) *V_T_* shifts to lower values (Figure 13a), raising the off-state current, and (ii) SCE becomes pronounced, degrading the subthreshold swing and DIBL. These detrimental consequences originate from a lesser gate control on both of the core (more distant) and the shell (as in thicker FD-SOI MOSFETs).

Figure 13b shows that the selection of the threshold voltage is a matter of trade-off between the thickness and the doping of the core.

Another important short-channel nuisance is the series resistance, discussed in the next section.

## 7. Back-Biasing

The biasing of the substrate or ground-plane is like adding a secondary gate to any SOI device. Back-biasing is instrumental for threshold voltage and mobility tuning, enhanced speed, and off-state power reduction [2,3]. Back-gate control is a privilege of FD-SOI, unheard of in FinFETs and GAA nanowires or nanosheets.

Dynamic tuning is also efficient, to some extent, in the CS-JL transistor. Figure 14a shows the lateral shift of subthreshold characteristics. A negative back-gate voltage *V_BG_* increases the threshold voltage as in an FD-SOI MOSFET but for a different reason. Since the bottom of the core becomes partially depleted, the effective doping is lower, which makes it easier for the front gate mission to fully deplete the core. A similar result is obtained by changing the ground-plane doping from N-type to P-type, as already noted in Figure 2. Conversely, a positive *V_BG_* produces carrier accumulation that enhances the effective doping and shifts the threshold voltage to negative values, as seen in Figure 9b. The transition from core conduction to shell activation is evidently extended.

The rate of change of threshold voltage in FD-SOI transistors depends essentially on the ratio between the thicknesses of the gate oxide and BOX [2,3,29]: Δ*V_T_*/Δ*V_BG_* ≈ *t_ox_*/*t_box_* = 70 mV/V. This relation also applies to the CS-JL FET, which is equally fully depleted in the subthreshold region.

Unlike FD-SOI, the back-gate action stops as soon as part of the core becomes neutral and screens the penetration of the vertical electric field from the substrate into the shell. Any variation in *V_BG_* is fully absorbed by the core and invisible to shell electrons. The threshold voltage of the shell and the carrier mobility cannot be adjusted by back-gate biasing any longer. This is why in on-state the transfer characteristics are rather immune to *V_BG_* (Figure 14b): the curves are shifted 10 times less than in weak inversion (Figure 14a). A higher core doping can completely isolate the shell. This screening mechanism is reciprocal to that produced by the inversion channel in the shell, which renders the core charge indifferent to front-gate voltage variations.

A negative *V_BG_* is effective in attenuating short-channel effects such as DIBL and subthreshold swing (Figure 15). The vertical field is stronger and pushes the carrier centroid to the front interface where the gate has plain control [30]. A fully depleted CS-JL transistor behaves as the FD-SOI counterpart.

The modulation of the electron concentration by the back-gate voltage affects not only the core but also the resistance of source and drain terminals. For negative *V_BG_* a depletion region develops from the core–BOX interface up according to doping concentration and bias (here limited to *|V_BG_|* ≤ 2 V): about 2 nm for 2 × 10^19^ cm^−3^ (see for example Figure 2b where *V_BG_* ≈ −1 V), 3 nm for 1 × 10^19^ cm^−3^ and 5 nm for 2 × 10^18^ cm^−3^.

The depletion effect lowers the total charge available in terminals and thus increases the series resistance *R_SD_*. Conversely, a positive *V_BG_* induces carrier accumulation over 2–3 nm depth, which reduces *R_SD_* (Figure 16). Note that the modulation of the series resistance is always profitable. A negative *V_BG_* is normally used in off-state mode to increase the threshold voltage and secure low leakage current; the increase in *R_SD_* helps. In on-state, the drive current is enhanced by lowering the threshold voltage with a positive *V_BG_*; the concomitant reduction of the series resistance is again a bonus.

The series resistance is usually extracted with the Y-function [26] or TLM method [31], which considers the total device resistance. *R_total_* adds the contributions of the channel R_ch_ and terminals *R_SD_*:(2)$$R_{total}=\frac{V_{D}}{I_{D}}=\frac{L}{\mu_{eff}WC_{ox}\left(V_{G}-V_{T}\right)}+R_{SD}$$

In principle, drawing *R_total_* versus channel length for various gate voltages results in straight lines that intercept at *L* = 0 and yield *R_SD_*. Unfortunately, the TLM method fails in CS-JL transistors where two parallel channels with distinct threshold voltages coexist. For the same reason, the Y-function cannot be applied either.

An intuitive appreciation of the series resistance is offered by the carrier distribution in the source (Figure 16a). A negative back-gate voltage can deplete most of the unraised region of the source which in turn increases considerably the series resistance. A positive V_BG_ induces a thin accumulation layer at the bottom of the source that lowers the series resistance. But, the effect of back bias is hardly visible in the raised section of the terminals.

The total charge available in the source and drain is computed in Figure 16b. The charge modulation via *V_BG_* is remarkable in devices without raised source and drain: 50% for 2 × 10^19^ cm^−3^ and 10-fold for 5 × 10^18^ cm^−3^ doping (from 1.8 × 10^−17^ to 1.7 × 10^−16^ C). The lower the doping concentration, the stronger the back-gate influence. However, in our typical CS-JL transistor with 15 nm raised terminals, the depletion or accumulation region is only a small fraction of the terminal thickness and is outweighed by the tall region, which is unaffected by *V_BG_*. The overall modulation effect is weak (Figure 16b): 13% for 2 × 10^19^ cm^−3^ and 50% for 5 × 10^18^ cm^−3^. In case of devices with highly doped terminals (>10^20^ cm^−3^), the back-gate effect is undetectable.

## 8. Conclusions

Compared with a conventional junctionless transistor, the superposition of an undoped layer (shell) on the core offers superior performance of CS-JL device in terms of drive current, transconductance, carrier mobility and threshold voltage tuning. Table 1 shows key metrics for CS-JL compared with conventional JL and FD-SOI transistors. The electron mobility in the shell is one order of magnitude higher than in the core and enables massively increased current. The size and doping of the core are key knobs for tuning the threshold voltage according to circuit requirements. With a thin core, the threshold voltage is positive, upgrading the device operation from normally-on to normally-off.

While the fundamental merit (no burden of junction fabrication) of junctionless MOSFET is preserved, its intrinsic inconveniences (poor mobility, normally-on operation) are eliminated. The planar CS-JL transistor takes advantage of back-biasing schemes, unavailable in nanowires and FinFETs.

The device architecture and processing are compatible with standard FDSOI process. The following are guidelines for optimization:(i)Keep ultrathin body (6−10 nm) as a safe protection against short-channel effects; this implies that both the core and the shell are 3−5 nm thick.(ii)Use as much core doping as possible to sustain low series resistance.(iii)Adjust core thickness to match the target threshold voltage.

Despite characteristics comparable with those of FD-SOI MOSFET, the CS-JL transistor is not meant to compete with. Its primary application is for 3D circuits and other devices that require low thermal budget.

## Figures and Tables

**Figure 1 micromachines-16-00418-f001:**
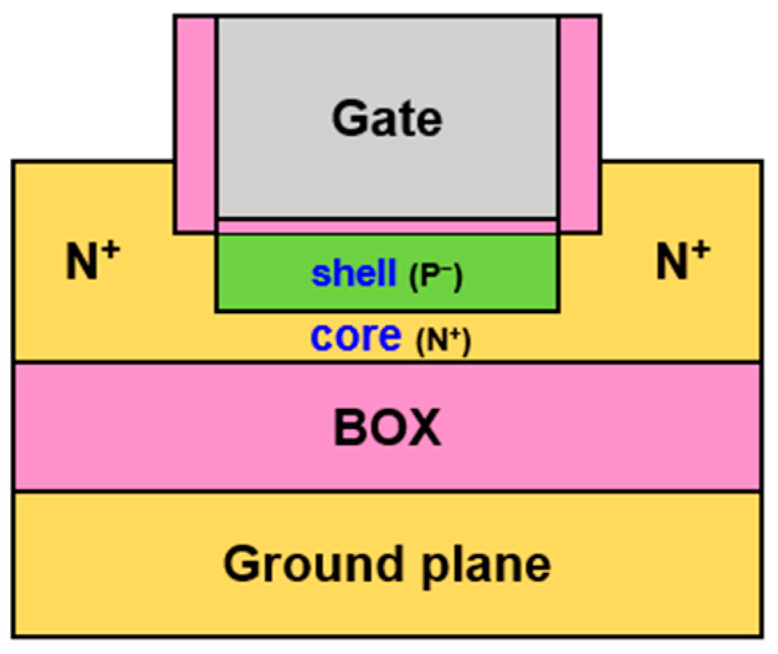
Schematic of planar core–shell junctionless transistor on FD-SOI platform.

**Figure 2 micromachines-16-00418-f002:**
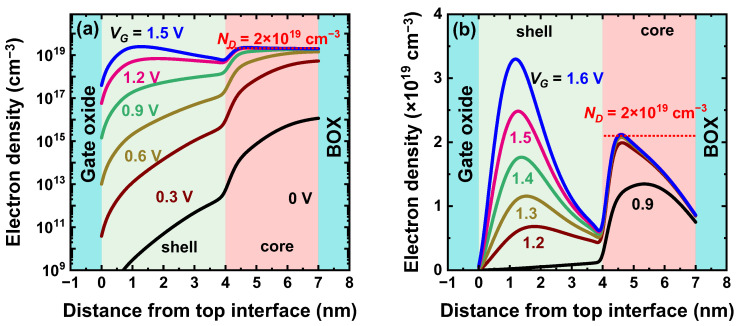
In-depth electron profiles illustrating the gradual filling of the core followed by the buildup of the inversion layer in the shell. N-channel CS-JL FET: 4 nm thick undoped shell, 3 nm thick core with *N_D_* = 2 × 10^19^ cm^−3^ doping, long gate (200 nm), 15 nm thick buried oxide, 1 nm thick gate oxide, low drain voltage (*V_D_* = 50 mV), and grounded back-gate. (**a**) N-type ground-plane, (**b**) P-type ground-plane, where the bottom of the core remains partially depleted. The simulations include quantum corrections.

**Figure 3 micromachines-16-00418-f003:**
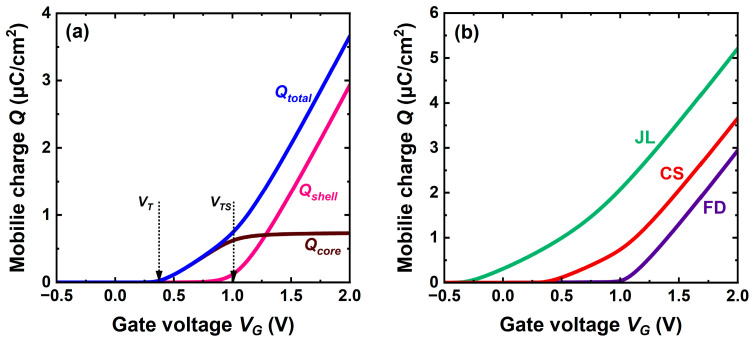
Mobile charge evolution with gate voltage. (**a**) Core filling followed by the buildup of the shell inversion charge. (**b**) Comparison of CS-JL, conventional JL, and FD-SOI MOSFETs, all with 7 nm thick bodies. CS-JL features a 4 nm thick undoped shell and a 3 nm thick core with *N_D_* = 2 × 10^19^ cm^−3^ doping (same doping for JL whereas FD-SOI is undoped). *L* = 200 nm and *V_D_* = 50 mV.

**Figure 4 micromachines-16-00418-f004:**
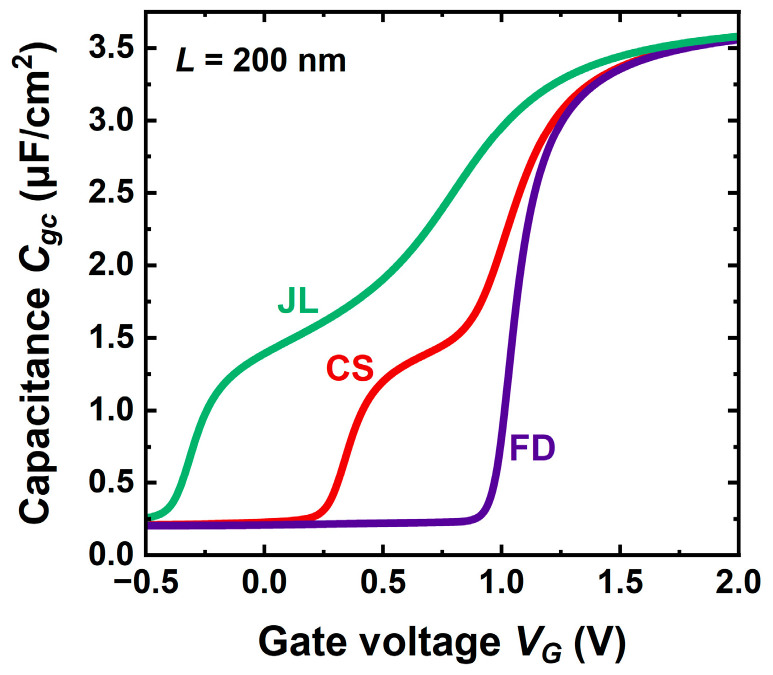
Gate-to-channel capacitance versus gate voltage in 7 nm thick JL, CS-JL, and FD-SOI transistors. Parameters as in Figure 3.

**Figure 5 micromachines-16-00418-f005:**
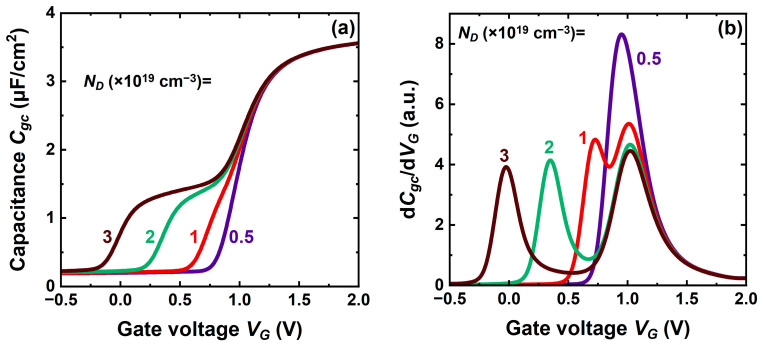
(**a**) Gate-to-channel capacitance versus gate voltage in CS-JL MOSFET with variable core doping levels. (**b**) Capacitance derivative shows the peaks corresponding to the threshold voltage of the core and shell. A 3 nm thick core and 4 nm thick shell, long gate (200 nm), 15 nm buried oxide, and 1 nm gate oxide.

**Figure 6 micromachines-16-00418-f006:**
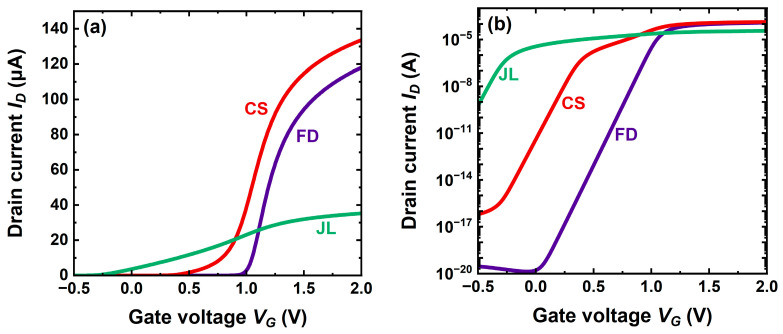
Transfer characteristics of planar CS-JL, conventional JL, and FD-SOI transistors, all with 7 nm thick bodies. (**a**) Linear and (**b**) semi-log scales. A 4 nm thick shell, 3 nm thick core with *N_D_* = 2 × 10^19^ cm^−3^ doping, long channel (200 nm), and low drain voltage (*V_D_* = 50 mV).

**Figure 7 micromachines-16-00418-f007:**
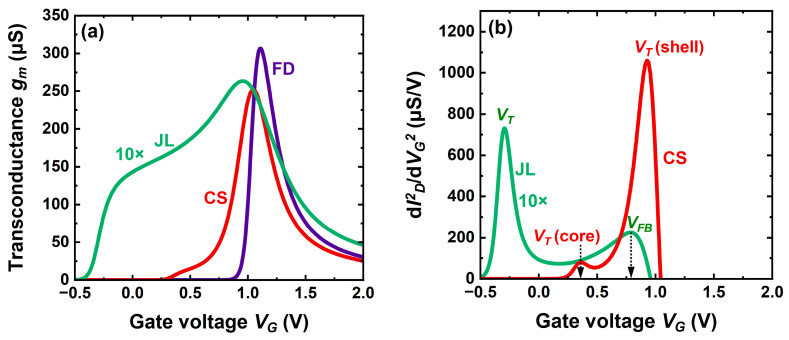
Transconductance (**a**) and its derivative (**b**) versus gate voltage in planar CS-JL, conventional JL, and FD-SOI transistors. The curves for JL device have been magnified 10-fold for visibility. Same parameters as in Figure 6.

**Figure 9 micromachines-16-00418-f009:**
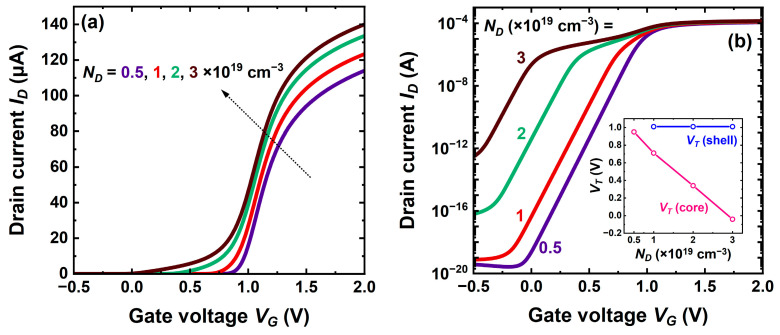
Transfer characteristics of planar CS-JL transistors with variable core doping levels. (**a**) Strong inversion and (**b**) subthreshold region; the inset shows the variation of the threshold voltage of core and shell with core doping. A 4 nm thick undoped shell, 3 nm thick core, *L* = 200 nm, and *V_D_* = 50 mV.

**Figure 10 micromachines-16-00418-f010:**
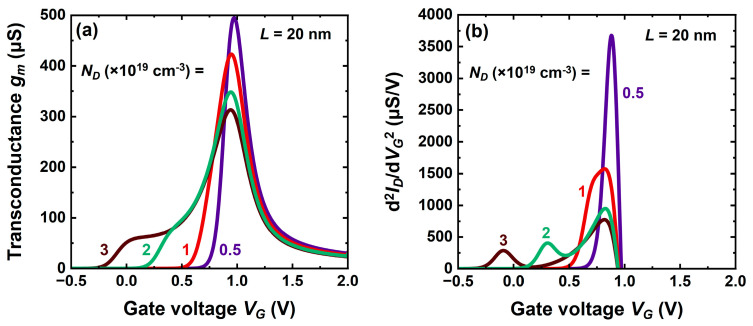
(**a**) Transconductance and (**b**) transconductance derivative of a CS-JL device with short channel *L* = 20 nm. A 4 nm thick undoped shell, 3 nm thick core, *L* = 200 nm, and *V_D_* = 50 mV.

**Figure 11 micromachines-16-00418-f011:**
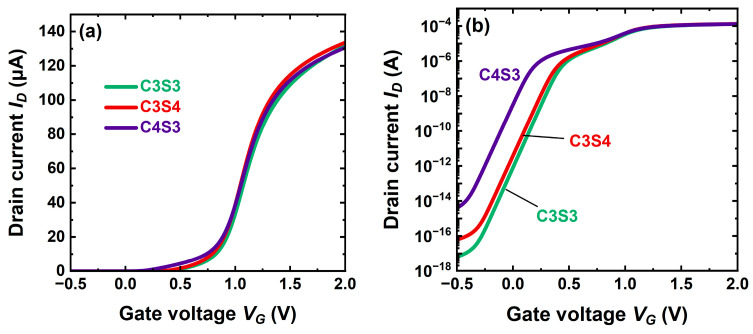
Transfer characteristics of planar CS-JL transistors with variable core and shell thicknesses: C4S3 stands for 4 nm core and 3 nm shell, C3S4 for 3 nm core and 4 nm shell, and C3S3 for 3 nm thick core and shell. (**a**) Strong inversion and (**b**) subthreshold region. A 2 × 10^19^ cm^−3^ core doping, *L* = 200 nm, and *V_D_* = 50 mV.

**Figure 12 micromachines-16-00418-f012:**
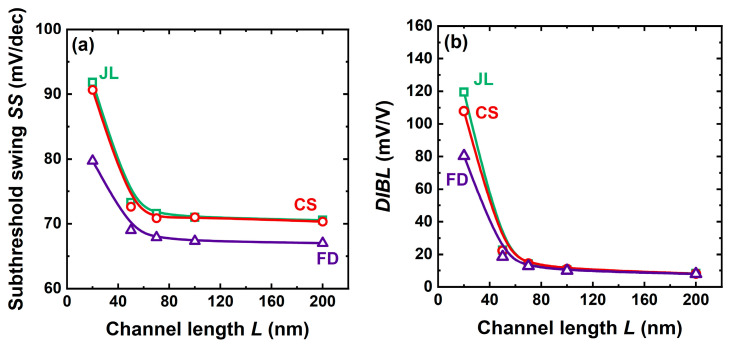
(**a**) Subthreshold swing and (**b**) DIBL as a function of channel length in CS-JL, FD-SOI, and JL transistors with 7 nm thick bodies. CS-JL features a 4 nm thick, undoped shell and a 3 nm thick core with *N_D_* = 2 × 10^19^ cm^−3^ doping (same doping for JL, whereas FD-SOI is undoped). *V_D_* = 50 mV.

**Figure 13 micromachines-16-00418-f013:**
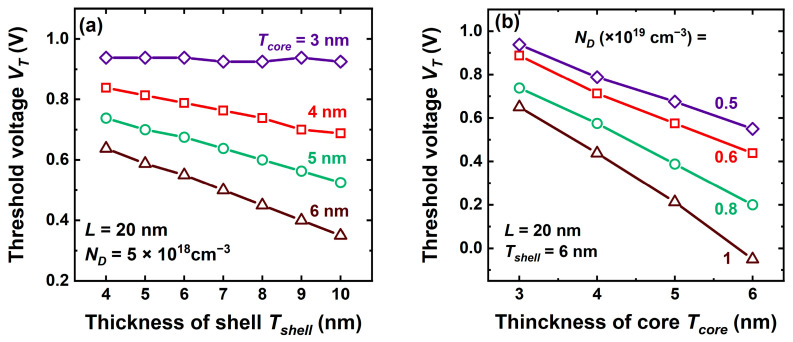
Threshold voltage variation with (**a**) shell and (**b**) core thickness in a very short CS-JL FET (*L* = 20 nm).

**Figure 14 micromachines-16-00418-f014:**
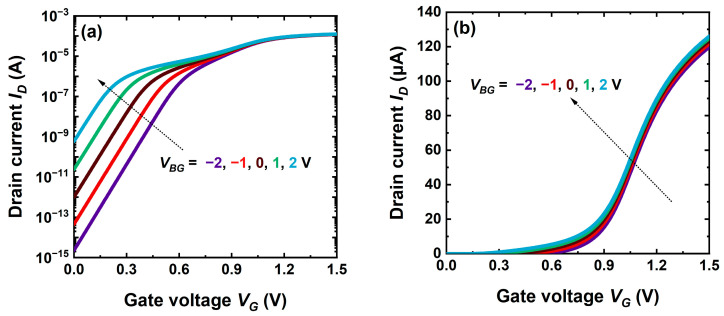
Impact of back-gate voltage on the transfer characteristics of planar CS-JL transistors in (**a**) weak and (**b**) strong inversion. A 4 nm shell and 3 nm core with 2 × 10^19^ cm^−3^ doping, *L* = 200 nm and *V_D_* = 50 mV.

**Figure 15 micromachines-16-00418-f015:**
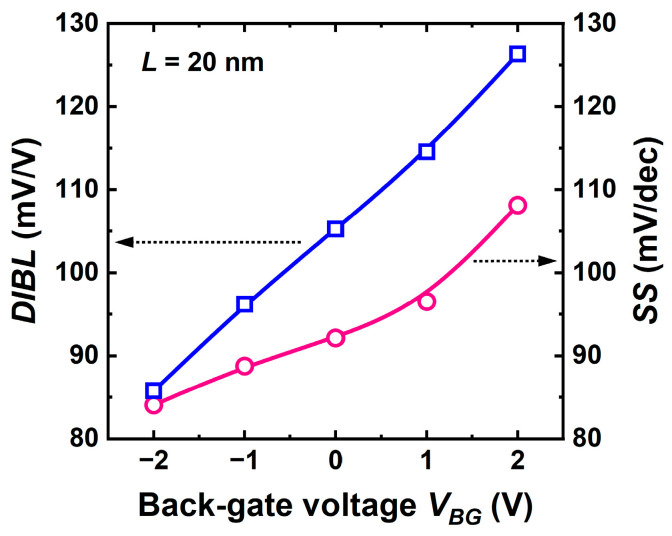
Subthreshold swing and DIBL versus back-gate voltage. Same parameters as in Figure 14 except for the channel length.

**Figure 16 micromachines-16-00418-f016:**
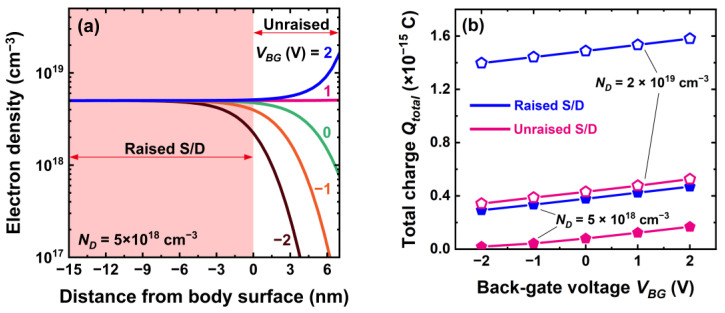
(**a**) In-depth profiles of electron concentration in the source terminal for variable back-gate voltage; the raised and unraised sections are indicated in colors. (**b**) Total charge in the source versus back-gate voltage at different core doping concentrations of CS-JL FETs with raised or unraised source and drain. A 4 nm shell and 3 nm core with 2 × 10^19^ cm^−3^ doping, *L* = 200 nm, and *V_D_* = 50 mV.

**Table 1 micromachines-16-00418-t001:** Key metrics for CS-JL, JL, and FD-SOI transistors.

	CS-JL	JL	FDSOI
Peak mobility (cm^2^/Vs)	640	50	700
Transconductance	high	poor	high
Drive current	high	poor	high
Threshold voltage	positive	negative	positive
S/D engineering	none	none	complex

## Data Availability

The data that support the findings of this study are available from the corresponding author upon reasonable request.

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
