# Peer review of "The Planar Core–Shell Junctionless MOSFET"

_micromachines, 2025, doi:10.3390/mi16040418_

Round 1

Reviewer 1 Report

Comments and Suggestions for Authors

The manuscript describes the fabrication and characterization of a core-shell junctionless (CS-JL) transistors, and compared the results against MOSFETs. The effects of back-gate voltage, core-doping and thickness on the electrical properties and performance of CS-JL transistors. Key findings include improved drive current, enhanced carrier mobility, and effective threshold voltage tuning. The CS-JL architecture is noted for its compatibility with standard processes and its potential for use in 3D circuits and low thermal budget applications, offering significant performance improvements.

This work is publishable with these questions addressed:

1) Are there any quantum tunneling effects or confinment effects from the QD? How would they change the performance of the device?

2) How does the core-shell architecture specifically contribute to the mitigation of short-channel effects compared to traditional junctionless and FD-SOI transistors? Also, how does the fringing electric field impact the device performance, and what improvements can be made, from your study?

3) What are the thermal characteristics of CS-JL FETs, and how do they compare to other transistor types? Electronic stability under multiple voltage cycles? This is related to the longevity of the devices. 

4) How compatible are CS-JL FETs with existing FD-SOI fabrication processes, and what modifications, if any, are necessary to integrate them into current manufacturing lines? Scalability of the device process is the question.

Comments on the Quality of English Language

The section on gate-to-channel capacitance characteristics introduces complex concepts that differ from regular MOSFETs. Simplifying the explanation or providing more context could help readers better understand.

 While the document examines short-channel effects, the technical details regarding the electric field distribution and its impact on device performance might be difficult for some readers to grasp without additional context or diagrams.

Author Response

Answers to Reviewers’ Comments

We thank you very much for taking the time to review our manuscript and make most valuable suggestions. Please find the detailed responses below and the corresponding corrections highlighted in the re-submitted file.

Reviewer 1

Comment 1:

Are there any quantum tunneling effects or confinement effects from the QD? How would they change the performance of the device?

Response 1:

We indeed considered the quantum effect. Figure 2 has been replaced with simulations showing the carrier profiles under quantum confinement.

Unlike the conventional FDSOI MOSFET, the CS-JL transistor is less affected: no visible effect on core transport and just a small increase of the threshold voltage of the shell.  Figure R1 shows the effect of quantum correction in TCAD simulations, where only the on-current is reduced by including the quantum confinement.

We do not elaborate on these secondary effects. The focus is to introduce a new device with extraordinary capability compared to the classical junctionless transistor. Our aim is to present the performance and main characteristics, by making clear the operating principles of CS-JL transistors, which are far more complex than in a MOSFET. Details on quantum and reliability effects, in particular at dimensional limits, will be addressed in subsequent papers.

Figure R1: Impact of confinement effects (at ND = 2×1019 cm−3) on the transfer characteristics of CS-JL transistor. The CS-JL devices are with 7 nm body (4 nm shell, 3 nm core) and at low drain voltage (VD = 50 mV). (a) Linear and (b) semi-log scales.

Corresponding change in manuscript: a sentence was added in section 2. Figure 2 and the caption have been modified.

Comment 2:

How does the core-shell architecture specifically contribute to the mitigation of short-channel effects compared to traditional junctionless and FD-SOI transistors? Also, how does the fringing electric field impact the device performance, and what improvements can be made, from your study?

Response 2:

The CS-JL transistor still belongs to junctionless family and maintains the advantage of junctionless in resisting the short-channel effect. However, a too thick shell compromises a little the SCE. The relevant results, including the fringing fields, are briefly discussed in Section 6 of the manuscript (Figures 11, 12).

Corresponding change in manuscript: sentence added and small corrections in section 6.

Comment 3:

What are the thermal characteristics of CS-JL FETs, and how do they compare to other transistor types? Electronic stability under multiple voltage cycles? This is related to the longevity of the devices.

Response 3:

Thank you for this pertinent suggestion for future work. We have not addressed thermal and reliability characteristics yet. A relatively thicker silicon film (core + shell) may contribute to power dissipation and alleviate thermal issue.

Corresponding change in manuscript: No

Comment 4:

How compatible are CS-JL FETs with existing FD-SOI fabrication processes, and what modifications, if any, are necessary to integrate them into current manufacturing lines? Scalability of the device process is the question.

Response 4:

The planar version of CS-JL transistors is process-compatible with FD-SOI. We are now pending several patents regarding the fabrication of CS-JL transistors on FD-SOI platform. The core-first and the shell-first options are mentioned at the end of Section 2.

Corresponding change in manuscript: No

Reviewer 2 Report

Comments and Suggestions for Authors

The manuscript presents a junction less MOSFETs with solid simulation-based evidence. The manuscript is well-organized and provides a thorough analysis of the CS-JL FET. However, some sections (e.g., Sections 3 and 5) could benefit from concise summaries at the end to reinforce key findings for readers. Addressing the above points—especially experimental validation and quantitative comparisons—would strengthen its impact. I recommend minor revisions before acceptance.

The planar CS-JL FET’s compatibility with FD-SOI is a strong point. Please clarify how this work distinguishes itself from prior CS-JL studies (e.g., nanowire versions in Refs. [8,9]) beyond the planar adaptation and the introduction would benefit from adding other examples for FETs such as doi.org/10.1002/anbr.202300055

The figures are informative, but captions could be more descriptive (e.g., Fig. 2 lacks details on simulation conditions). Please enhance captions for standalone comprehension.

In the Abstract, the authors claim "one order of magnitude higher mobility" due to the shell. Can you quantify this (e.g., peak mobility values) in the abstract for clarity?

In Section 2 (Architecture), The core-first vs. shell-first fabrication processes are intriguing. Could you briefly discuss trade-offs (e.g., cost, scalability) between these approaches?

In Section 3 (Charge Distribution), The two-phase charge buildup is well-explained, but the impact of P-type vs. N-type ground-planes (Fig. 2) on core depletion is underexplored. Could you elaborate on how ground-plane doping affects performance?

In Section 4 (Capacitance), The "shoulder" in capacitance curves is a key feature. Can you provide experimental data (beyond simulations) to validate this, given A2RAM’s experimental success (Refs. [15-17])?

In Section 5B (Mobility), The mobility extraction using the Y-function assumes a constant core current post-shell activation, yet the authors note this is an approximation (p. 8). Could you quantify the error introduced by this assumption?

In Section 6 (Short-Channel Effects), The resilience of CS-JL to SCE is promising, but the comparison with JL and FD-SOI (Fig. 12) lacks discussion on DIBL’s physical origins. Please clarify the role of the shell in mitigating SCE.

In Section 7 (Back-Biasing), The screening effect of the core is a critical insight. Can you provide a quantitative metric to support this claim?

In Conclusion, The claim of "superior performance" is strong. Could you add a table summarizing key metrics for CS-JL vs. JL vs. FD-SOI to solidify this?

Eq. (1) (p. 8) lacks a definition of W and L. Please define all variables explicitly.

Typos: Minor errors, e.g., "oustate" (p. 12, line 351) should be "off-state." Please proofread.

Author Response

Answers to Reviewers’ Comments

We thank you very much for taking the time to review our manuscript and make most valuable suggestions. Please find the detailed responses below and the corresponding corrections highlighted in the re-submitted file.

Reviewer 2

Comment 1:

The planar CS-JL FET’s compatibility with FD-SOI is a strong point. Please clarify how this work distinguishes itself from prior CS-JL studies (e.g., nanowire versions in Refs. [8,9]) beyond the planar adaptation and the introduction would benefit from adding other examples for FETs such as doi.org/10.1002/anbr.202300055.

Response 1:

The nanowire version could deliver superior device performance at advanced nodes but suffers from complex fabrication process. The Gate-All-Around architecture is ideal for electrostatic control but hard to implement together with the core-shell structure. This is why we consider the pragmatical solution of planar CS-JL on FD-SOI platform. FD-SOI is known for high RF performance, reliability, energy saving and wide temperature range. We believe that the FD-SOI-based planar CS-JL FET has many opportunities in those fields. Note that the simple fabrication with low thermal budget will boost 3D sequential integration of CS-JL FETs by using FD-SOI technology. Medical applications like in the “doi” you kindly suggested are a far shot.

Corresponding change in manuscript: the third paragraph in Introduction has been modified for clarity.

Comment 2:

The figures are informative, but captions could be more descriptive (e.g., Fig. 2 lacks details on simulation conditions). Please enhance captions for standalone comprehension.

Response 2:

All figure captions have been revised and completed.

Comment 3:

In the Abstract, the authors claim "one order of magnitude higher mobility" due to the shell. Can you quantify this (e.g., peak mobility values) in the abstract for clarity?

Response 3:

The “peak mobility” was added in the abstract.

Comment 4:

In Section 2 (Architecture), The core-first vs. shell-first fabrication processes are intriguing. Could you briefly discuss trade-offs (e.g., cost, scalability) between these approaches?

Response 4:

We prefer to keep the generic description of the two ways of fabrication, without more details, because several patents are pending. Thank you for your understanding.

Comment 5:

In Section 3 (Charge Distribution), The two-phase charge buildup is well-explained, but the impact of P-type vs. N-type ground-planes (Fig. 2) on core depletion is underexplored. Could you elaborate on how ground-plane doping affects performance?

Response 5:

A clarifying sentence was added below Fig. 2 and another in section 7.

Comment 6:

In Section 4 (Capacitance), The "shoulder" in capacitance curves is a key feature. Can you provide experimental data (beyond simulations) to validate this, given A2RAM’s experimental success (Refs. [1-17])?

Response 6:

Excellent hint, thank you. Experimental CV curves with plateau and double-peak derivative have been reported in A2RAM.

Corresponding change in manuscript: several sentences have been included in Section 4 and reference 20 has been updated.

Comment 7:

In Section 5B (Mobility), The mobility extraction using the Y-function assumes a constant core current post-shell activation, yet the authors note this is an approximation (p. 8). Could you quantify the error introduced by this assumption?

Response 7:

Indeed, the Y-function implementation was based on the assumption of constant core current after shell activation.  To address this approximation, we have computed the core and shell currents separately and isolated the shell current from the total current. The extracted low-field mobility is higher compared to that delivered by the approximation of constant core current, see figure below. Note that the error can differ as the configuration changes, e.g., thickness, doping, and channel length. The computation of the exact shell current is feasible in TCAD only. In practice, the Y-function is still useful to extract the mobility. Even with an underestimated value, it shows the outstanding difference between core and shell mobilities.

Figure R3: The implementation of Y-function. (Left) Ishell extracted from TCAD and (Right) by using assumption.

Corresponding change in manuscript: the paragraph before Fig. 8 has been updated and enriched.

Comment 8:

In Section 6 (Short-Channel Effects), The resilience of CS-JL to SCE is promising, but the comparison with JL and FD-SOI (Fig. 12) lacks discussion on DIBL’s physical origins. Please clarify the role of the shell in mitigating SCE.

Response 8:

Thanks for this suggestion.

In on-state, CS-JL transistor behaves like FD-SOI, so the impact of thickness on SCE is similar. Specifically, the thicker the shell, the stronger the SCE, manifesting as degraded DIBL, SS, and Vt roll-off. The presence of the core degrades the short-channel characteristics. As a result, the overall SCE of CS-JL is in-between FD-SOI and JL.

Corresponding change in manuscript: the text above Fig. 12 has been revised.

Comment 9:

In Section 7 (Back-Biasing), The screening effect of the core is a critical insight. Can you provide a quantitative metric to support this claim?

Response 9: 

The higher the doping, the stronger the screening effect.

Corresponding change in manuscript: the text before Fig. 14 has been enriched.

Comment 10:

In Conclusion, the claim of "superior performance" is strong. Could you add a table summarizing key metrics for CS-JL vs. JL vs. FD-SOI to solidify this?

Response 10:

A Table was added to show the superior performance of CS-JL compared with classical JL transistor.

Comment 11:

Eq. (1) (p. 8) lacks a definition of W and L. Please define all variables explicitly.

Response 11:

These variables are now defined below Eq. 1.

Comment 12:

Typos: Minor errors, e.g., "oustate" (p. 12, line 351) should be "off-state." Please proofread.

Response 12:

Corrected, thank you for very careful reading.
